# SIK2 Controls the Homeostatic Character of the POMC Secretome Acutely in Response to Pharmacological ER Stress Induction

**DOI:** 10.3390/cells13181565

**Published:** 2024-09-17

**Authors:** Mehmet Soner Türküner, Ayşe Yazıcı, Ferruh Özcan

**Affiliations:** 1Department of Molecular Biology and Genetics, Graduate School of Natural and Applied Sciences, Gebze Technical University (GTU), Gebze, Kocaeli 41400, Turkey; msturkuner@gtu.edu.tr (M.S.T.); ayseyazici@gtu.edu.tr (A.Y.); 2Cellular Proteomics Laboratory, Gebze Technical University—Central Research Laboratory, Application and Research Center Laboratory (GTU-MAR), Gebze, Kocaeli 41400, Turkey

**Keywords:** SIK2, POMC, inflammation, neuronal secretome, ER stress, hypothalamic obesity

## Abstract

The neuronal etiology of obesity is centered around a diet-induced inflammatory state in the arcuate nucleus of the hypothalamus, which impairs the functionality of pro-opiomelanocortine neurons (POMCs) responsible for whole-body energy homeostasis and feeding behavior. Intriguingly, systemic salt inducible kinase 2 (SIK2) knockout mice demonstrated reduced food intake and energy expenditure along with modestly dysregulated metabolic parameters, suggesting a causal link between the absence of SIK2 activity in POMCs and the observed phenotype. To test this hypothesis, we conducted a comparative secretomics study from POMC neurons following pharmacologically induced endoplasmic reticulum (ER) stress induction, a hallmark of metabolic inflammation and POMC dysregulation in diet-induced obese (DIO) mice. Our data provide significant in vitro evidence for the POMC-specific SIK2 activity in controlling energy metabolism and feeding in DIO mice by regulating the nature of the related POMC secretome. Our data also suggest that under physiological stress conditions, SIK2 may act as a gatekeeper for the secreted inflammatory factors and signaling molecules critical for cellular survival and energy homeostasis. On the other hand, in the absence of SIK2, the gate opens, leading to a surge of inflammatory cytokines and apoptotic cues concomitant with the dysregulation of POMC neurons.

## 1. Introduction

A paradigm shift in the etiology of obesity occurred when it became clear that low-grade metabolic inflammation was detected in several brain regions, particularly in the arcuate nucleus (ARC) of the hypothalamus, even before the onset of obesity. The ARC of the hypothalamus is a key regulatory center for whole-body energy homeostasis and feeding behavior and transmits signals between the periphery and the central nervous system (CNS) [1]. The two functionally opposing neuronal subpopulations consisting of pro-opiomelanocortin (POMC) and neuropeptide Y/agouti-related protein (NPY/AgRP) neurons in the ARC integrate peripheral and central inputs and communicate with their downstream effectors through various secreted proteins for the maintenance of organismal energy balance [1,2]. In the low-energy state, some gut-derived hormones, such as ghrelin, activate orexigenic NPY/AgRP neurons, which stimulate food intake and suppress energy expenditure [3]. In contrast, in the excess energy state, nutrient-related hormones, such as leptin and insulin, activate anorexigenic POMC neurons, which enhance energy expenditure while decreasing food intake [4,5]. POMC, which is expressed as a precursor, undergoes a series of posttranslational cleavages, resulting in the secretion of biologically active neuropeptides (such as α/β-melanocyte-stimulating hormone (α/β-MSH), adrenocorticotropic hormone (ACTH), and β-endorphin) [6] in addition to fast neurotransmitters such as γ-aminobutyric acid (GABA) [7]. These neuropeptides play important roles in various physiological processes, including microglial inflammation [8], pigmentation [9], stress response, and energy balance [10,11]. POMC neurons project specifically to the paraventricular (PVH) and dorsomedial (DMH) nuclei of the hypothalamus, the lateral hypothalamic area (LHA) [7,12] and the medial amygdala [13]. POMC neuron-derived neuropeptide α-MSH bind to their cognate receptor, melanocortin 4 receptor (MC4R), on PVN neurons, thereby inducing signaling cascades to maintain energy homeostasis.

Salt inducible kinase 2 (SIK2), a member of the AMP-activated protein kinase (AMPK) family, is highly expressed in adipose tissue and the liver [14] and plays a pivotal role in glucose and lipid metabolism [15]. Previous studies have emphasized the possible modulatory functions of SIK2 in neuronal survival [16] and proteinopathy-related diseases [17]. SIK2 reduces the nuclear translocation of the target of rapamycin (TOR) complex-2 (TORC2) and corticotropin-releasing hormone (CRH) transcription in the hypothalamic neurons of rats in a stress-dependent manner [18]. SIK2 contributes to insulin secretion in pancreatic beta cells by promoting calcium influx via voltage-dependent calcium channels under hyperglycemic conditions [19]. SIK2 and SIK3 were implicated in macrophage polarization and secretion of inflammatory cytokines in monocytes and must cells, respectively [20,21]. Intriguingly, SIK2 localizes mainly to the ER membrane and undergoes rapid transient activation upon ER stress induction independent of its upstream activator kinase, liver kinase B1 (LKB1). Although the functional importance of its ER localization and the identity of its upstream activator kinase are not fully understood, a recent study revealed that the UPR is involved in the ER stress-dependent activation of SIK2 [22]. Moreover, SIK2 has been repeatedly implicated in the regulation of major ER stress response pathways such as inflammation, ERAD, and autophagy downstream of UPR activation. SIK2 mediates the upregulation of ERAD and autophagy flux by phosphorylating p97/VCP [23], an unidentified factor involved in autophagosome–lysosome fusion, respectively [17], on the ER membrane. Autophagosomes are known to participate in the secretion of leaderless proteins, although the molecular mechanisms related to vesicular diversification between autophagosomes and the secretion of vesicles have yet to be elucidated. Consistent with these findings, a recent report proposed that SIK2 enrichment in ER lipid rafts is required for the proper packaging of secretory cargo proteins into export vesicles, suggesting a molecular mechanism that enables SIK2 to modify the level and/or identity of secreted proteins. In the same study, SIK2 was also shown to relocate to a distinct ER subdomain to promote autophagy and protect cells from apoptosis under ER stress conditions [24].

The total secretome, a collection of proteins secreted or shed by cells, plays a central role in many biological processes and shapes the microenvironment of cells [25]. The content of the neuronal secretome plays a crucial role in neuroprotection, neurodegeneration, and signal transmission in health and neurodegenerative diseases, underscoring its therapeutic potential [26,27,28]. Hypothalamic inflammation is characterized by an abrupt surge of cytokines and chemokines, including fractalkine (CXC3CL1), monocyte chemoattractant protein-1 (MCP-1), tumor necrosis factor-α (TNF-α), interleukin-6 (IL-6), and a few others, which act on POMC neurons by altering their secreted neuropeptides and neurotransmitters, which further aggravates the inflammatory state culminating in POMC-mediated dysregulation of food intake and energy expenditure [29,30]. Although controversial data exist for the cellular origin and significance of hypothalamic inflammation in obesity development, the future design of centrally acting anti-obesity drugs will most likely target the specific links between the neurons and non-neuronal components of the feeding circuitry via their secreted neuropeptides and/or neurotransmitters.

Earlier reports have demonstrated that low-grade metabolic inflammation in the hypothalamic arcuate nucleus of obese animal models can alter the expression of anorectic POMC and, thereby, its derivative neuropeptides, altering the feeding behavior and energy metabolism of the organism [31]. In vivo studies with obese animal models (DIO) have demonstrated a correlation between hypothalamic inflammation and ER stress and the lipid content of a high-fat diet (HFD) [32]. In our in vitro model system, we primarily aimed at probing the effects of chemically induced ER stress and its downstream effector kinase SIK2 on the POMC secretome to avoid compounding effects associated with the use of physiologically relevant saturated fatty acids such as palmitate [33]. In this study, our strategy was to elucidate SIK2-mediated changes in the ER-stressed POMC secretome that may contribute to disease etiopathogenesis. In the end, these neuropeptides may provide therapeutically invaluable targets for the treatment of obesity and associated diseases.

## 2. Materials and Methods

### 2.1. Cell Culture and Chemical Treatments

The mouse hypothalamic proopiomelanocortin cell line *mHypoA-POMC/GFP-2* (POMCs) was initially derived from the arcuate nucleus of the mouse, expressing the enhanced green fluorescent protein (EGFP) transgene driven by the POMC promoter through the use of fluorescence-activated cell sorting (FACS). It has been previously demonstrated that these cells secrete α-MSH and, consistent with their origin, respond to anorectic hormones such as insulin and leptin as well as the ER stressor palmitate. Consequently, it has been reported that these cells can be used as accurate in vitro models for the discovery, development, and validation of novel therapeutics targeting central nervous system-based diseases and disorders, including obesity, stress, and metabolic disorders [34,35].

POMCs and human embryonic kidney (HEK) 293FT cells were purchased from CEDARLANE (Burlington, ON, Canada) and Invitrogen (Waltham, MA, USA), respectively. The cells were maintained in complete high-glucose Dulbecco’s Modified Eagle Medium (DMEM) (Gibco, Waltham, MA, USA) supplemented with 10% fetal bovine serum (FBS) (Gibco) and 1% penicillin/streptomycin (Gibco) at 37 °C and 5% CO_2_. The cells were cultured to 70–80% confluence and passaged as suggested by the vendor.

The thapsigargin (THA) (Sigma Aldrich, Saint Louis, MO, USA), which is a chemical agent causing calcium depletion in the ER by inhibiting sarco/endoplasmic reticulum Ca^2+^ transport ATPase (SERCA), was used as an ER stress inducer. It was dissolved in dimethyl sulfoxide (DMSO) (Sigma Aldrich), which was used as a control group in the experiments. ER stress induction with THA (100 nM) [36] was verified by an increase in the level of the luminal chaperone glucose-regulated protein 78 (GRP78/BiP) by immunoblotting (CST, Danvers, MA, USA).

### 2.2. Cell Viability Assays

To investigate the cytotoxicity of THA on POMCs, 5000 cells were seeded in each well of a 96-well plate. After 24 h, the cells were treated with 100 nM THA and vehicle (DMSO) for the indicated durations (6, 12, and 24 h). In the last 4 h of treatment, 10 µL of the cell proliferation reagent water-soluble tetrazolium salt (WST-1) (Roche, Basel, Switzerland) was added to the wells. The cells were incubated for 4 h at 37 °C with 5% CO_2_. Spectrophotometric measurements were performed at 450 nm (with 600 nm as a reference wavelength) by using a Varioskan Flash Reader (Thermo Scientific, Waltham, MA, USA).

To examine the viability of POMCs under THA and serum-free conditions, 25,000 cells were seeded in a 24-well plate. After 24 h, under serum-depleted conditions, the cells were washed 3 times with 1× phosphate-buffered saline (PBS) (LONZA, Basel, Switzerland) and then treated with 100 nM THA and vehicle DMSO in 0% serum-containing media for 6 and 12 h. For serum-containing conditions, treatment was performed directly in the supplemented media. The percentage of dead cells was quantified by counting viable cells with trypan blue under a DMi1 light microscope (Leica, Nußloch, Germany).

### 2.3. Preparation of Molecular Cloning and Manipulation Cell Lines

#### 2.3.1. Transient Overexpression System

Wild-type (WT) SIK2 was subcloned and inserted into the pCMV-3Tag-6 expression vector (#240200 Agilent, Santa Clara, CA, USA) from the pDream_mSIK2 WT plasmid containing the synthetic mouse SIK2 gene (Genescript, Piscataway, NJ, USA). For this cloning, a proper primer pair was designed by using the cloud-based platform Benchling (San Francisco, CA, USA) and Oligo Analyzer tool (IDT, Coralville, IA, USA), and then the primer pair was synthesized by Macrogen (Seoul, Republic of Korea) (Table 1). The gene was inserted from donor plasmids by using high-fidelity PrimeStar HS DNA Polymerase (Takara, San Jose, CA, USA). The produced insert and targeted plasmid were cut by fast digest restriction enzymes (Thermo Scientific, USA) and extracted from a 1% agarose gel by using an MN Gel Extraction or PCR Clean-Up Kit (Macherey-Nagel, Düren, Germany). Ligation of the prepared insert and plasmid via T4 DNA Ligase (NEB, USA) and transformation of the ligation products into competent DH5α cells were performed via the conventional transformation method (Addgene, Watertown, MA, USA). After obtaining colonies, colony PCRs were performed with the designed primer pair and Taq DNA Polymerase (ABM, Richmond, BC, Canada) to identify possible positive colonies. To further confirm the presence of the insert within the recombinants, plasmids were isolated from the detected colonies by a DNA-spin Plasmid DNA Purification Kit (iNtRON Biotechnology, Seongnam-si, Republic of Korea) and were reconstituted with the same restriction enzymes that were used during the cloning step to observe the inserts. At least three of the confirmed colonies were selected and sequenced (Macrogen) to perform the final control. Finally, the pCMV-3Tag6-mSIK2_WT plasmid was isolated using the MN NucleoBond Xtra Midi Kit to obtain transfection-grade plasmids.

To generate flag-tagged SIK2 WT overexpressing and control POMC cells, the pCMV-3Tag6-mSIK2_WT plasmid and empty pCMV-3tag6 plasmid were transfected into POMCs, respectively, using polyethyleneimine (PEI-MAX) (Polysciences, Warrington, PA, USA). To increase the transfection efficiency, a series of optimizations of salt ions and related parameters that affect PEI-DNA particles were performed based on a previous report [37]. A total of 800,000 cells were seeded in 10 cm cell culture dishes. After 24 h, 15 µg of plasmid DNA and PEI were incubated in 1× Tris/EDTA buffer containing 150 mM NaCl (pH 8.0) at a ratio of 1:3 (*w*:*w*) for 10 min, after which the PEI: DNA particles were distributed throughout the cells. After 6 h, the media of the cells were replaced with fresh, complete DMEM. Transiently transfected POMC cells were used for experiments at 48–72 h post-transfection.

#### 2.3.2. Lentivirus-Based Stable shRNA System

For RNA interference-based SIK2 knockdown (KD), three different shRNA oligos that target endogenous SIK2 mRNA, which is complementary to the indicated exon positions, were designed by using the siRNA selection tool WI siRNA Selection Program (MIT, Cambridge, MA, USA) based on the design rules of the Broad Institute [38] (Table 1). Synthesized oligos from Macrogen were cloned and inserted into the lentiviral pLKO.1 TRC cloning vector (#10878 Addgene), which was used according to the suggested protocols [39]. At least three confirmed clone colonies were selected and sent to Macrogen for sequencing. The confirmed plasmids were isolated using an MN NucleoBond Xtra Midi Kit to obtain transfection-grade plasmids.

To produce lentiviral particles, one day before transfection, 293FT cells were seeded in 6 cm cell culture dishes at 80–90% confluence on the day of transfection. Lentiviral packaging was performed by transferring 2nd Generation Packaging System from Addgene; 3 µg of psPAX2 (#12260), 2 µg of pCMV-VSVG (#84549), and 4 µg of cloned pLKO.1-TRC lentiviral expression vectors into 293FT cells by PEI-based transfection (as described previously). Following a 48 h and 72 h period of transfection, the media containing viral particles were harvested via centrifugation (1500 rpm for 5 min) and then filtered (0.45 μm filter unit attached to a syringe) from the 293FT cells. The viral particles were aliquoted for single use at −80 °C. To generate stable SIK2-KD and scrambled POMC cell lines, 100,000 cells were seeded in a 6-well plate. After 24 h, lentiviral transduction was performed using 8 µg/mL of the cationic polymer hexadimethrine bromide (polybrene) (Sigma Aldrich). Stable cells were obtained following selection with 5 µg/mL puromycin (Santa Cruz Biotech, Dallas, TX, USA) for 14 days. Stable POMCs were tested for their knockdown efficiency, and stable scramble-POMCs were used for the RNAi control group.

### 2.4. Conditioned Media (CM) Isolation

To obtain a pure secretome sample, our protocol was optimized according to previous reports [40]. A total of 800,000 POMCs were seeded in 10 cm cell culture dishes. After 24 h, the cells were washed 3 times with 1XPBS (LONZA) and then treated with 100 nM THA and vehicle DMSO in 5 mL of 0% serum-containing DMEM for 6 and 12 h. The CMs collected from three 10 cm cell culture dishes per condition were pooled. Fifteen milliliters of medium from each condition was centrifuged at 500× *g* for 5 min and then at 2000× *g* for 10 min at 4 °C, and the resulting supernatant was passed through a 0.22 µm polyethersulfone (PES) filter to remove cellular debris and apoptotic bodies. The secretome samples were centrifuged at 10,000× *g* for 30 min at 4 °C to remove microvesicles and then centrifuged at 100,000× *g* for 1 h at 4 °C to remove exosome vesicles. After all these serial centrifugation processes, the resulting pure secretome samples were concentrated to approximately 200–500 µL and buffer-exchanged into 50 mM Tris, pH 8.0, using an Amicon^®^ Ultra Centrifugal Filter with a 3 kDa molecular weight cutoff (MWCO) (Merck, Rahway, NJ, USA). The total protein concentration was determined using a Qubit 3.0 with a Qubit Protein Assay Kit (Thermo Scientific).

### 2.5. Immunoblotting and One-Dimensional Gel Electrophoresis (1DE) Analysis

The total protein for each condition was isolated in lysis buffer (150 mM NaCl, 50 mM Tris-Cl (pH 7.4), 2 mM EDTA, 1% NP-40, 0.5% sodium deoxycholate, 5 mM sodium fluoride (NaF), 1 mM sodium orthovanadate (Na_3_VO_4_), 1 mM phenylmethylsulfonyl fluoride (PMSF), 5 mM aprotinin, and 1 μg/mL leupeptin) (Applichem, Darmstadt, Germany) by centrifugation at 18,000× *g* for 15 min at 4 °C. The total protein concentration was quantified with a Pierce Bicinchoninic acid (BCA) Protein Assay Kit (Thermo Scientific). The samples were loaded equally on 5% and 10% sodium dodecyl-sulfate polyacrylamide gel electrophoresis (SDS–PAGE) gels and run at 80 volts. Proteins in 5% gels were transferred to polyvinylidene fluoride (PVDF) membranes (Thermo Scientific) by Transblot Turbo according to the manufacturer’s instructions (Bio-Rad, Hercules, CA, USA), while those in 10% gels were subjected to SybroRuby Gel Stain (Invitrogen) for total protein staining according to the manufacturer’s instructions. The membranes were blocked in 5% skim milk in Tris-buffered saline (TBS) containing 0.1% Tween-20 for 1 h at room temperature and immunoblotted with the following primary antibodies: anti-SIK2 (1:1000; Cell Signaling; #6919), anti-p-SIK2 (1:1000; Invitrogen; #PA5-64607), anti-BiP (1 :1000; Cell Signaling; #3177), anti-α-Tubulin (1:1000; Cell Signaling; #3873), and anti-β-Actin (1:1000; Cell Signaling; #4970) overnight at 4 °C and with the following secondary antibodies: anti-mouse IgG-HRP (1:3000, Santa Cruz, #sc-2005) and anti-rabbit IgG-HRP (1:3000, Santa Cruz, #sc-2357) for 2 h at room temperature. Immunoblots and gel staining were visualized with ChemiDoc XRS+ (Bio-Rad). Chemiluminescence-derived bands were detected by WesternBright enhanced chemiluminescence horseradish peroxidase (ECL HRP) substrate (Advansta, San Jose, CA, USA) and quantified using ImageJ software (v.1.48) as previously reported [41].

### 2.6. Secretome Analysis

#### 2.6.1. Sample Preparation

Five micrograms of each sample were dissolved in 50 mM Tris-HCL, pH 8.0 (Invitrogen), containing 6 M urea (Invitrogen) for full denaturation. In a thermoshaker (400–500 rpm), the proteins were reduced with 500 mM tris (2-carboxyethyl) phosphine (TCEP) (Thermo Scientific) (20 mM final concentration) for 1 h at 60 °C, followed by alkylation using 1 M iodoacetamide (IAA) (Thermo Scientific) (40 mM final concentration) for 30 min at room temperature. After dilution (urea < 1 M) with 50 mM Tris-HCl (pH 8.0), the proteins were digested with Pierce trypsin protease (Thermo Scientific) (1:20, *w*:*w*) overnight at 37 °C in a thermoshaker. To stop the digestion, formic acid (1–2% final concentration) was added to each sample. The peptides were concentrated and desalted by a Pierce C18 tip (Thermo Scientific) and speed-vac.

#### 2.6.2. Mass Spectrometry

The desalted and concentrated peptides were loaded onto a reverse-phase column on a nanoflow ultrahigh-performance liquid chromatography (nano-UHPLC) instrument (Ultimate 3000 RSLCnano, Thermo Scientific) coupled online to an LTQ Orbitrap ELITE mass spectrometer with a nanoelectrospray ion (ESI) source (Thermo Scientific). The peptides were mixed to 200 ng/µL with a sampler buffer containing 5% acetonitrile and 0.1% formic acid and were introduced into the liquid chromatograph with a 2 µL injection volume. Mobile phase A was 0.1% (*v*/*v*) formic acid, and mobile phase B was 0.1% (*v*/*v*) formic acid in 80% acetonitrile (Thermo Scientific). Peptide gradient separation was performed on a C18 column (Acclaim PepMap, RSLC 75 µm × 250 mm) (Thermo Scientific). Peptide separation was conducted using the following gradients: 4% phase B over 5 min, 4–55% phase B over 85 min, 55% phase B over 10 min, and 55–95% phase B over 10 min, with a final elution of 95% B for 5 min at a flow rate of 300 nL/min. Each sample was analyzed by LC–MS/MS as three fractions. The data were acquired using the data-dependent acquisition (DDA) method in positive ion mode. MS1/MS2 scans were performed in *Orbitrap* and *ion trap* analyzers, respectively, with 350–1800 m/z scan ranges at 60,000 resolution. All MSn spectra were acquired using collision-induced dissociation (CID) with 35% normalized collision energy, 0.25 activation Q, a 10 ms activation time, and a 1 m/z isolation width. The data were acquired using Xcalibur 4.0 software (Thermo Scientific). The raw MS data were analyzed using Proteome Discoverer 2.3 software (PD) (Thermo Scientific).

### 2.7. Bioinformatic and Statistical Analysis

The results of the LC-MS/MS analysis for all samples are based on independent biological triplicates and three technical fractions for each experimental condition (*n* = 9). MS/MS spectra were searched and identified against the mouse total protein database, secretome database (“Secreted” from the cellular component of Gene Ontology), and sheddome database (“Transmembrane” and “GPI anchored”) from UniProt by using Sequest HT and msAmanda 2.0 search engines. The tolerances of the precursor and fragment ions were 10 ppm and 0.5 Da, respectively. Oxidation (M), N-terminal acetylation, and carbamidomethylation were selected as the dynamic and static modifications, respectively. Peptide spectrum matches (PSM) validations were performed by “Percolator” using decoy database search with two false discovery rate (FDR) thresholds: strict FDR set to 0.01 and relaxed FDR set to 0.05. As a measure of confidence in peptide identification, those with an FDR above 0.05 are classified as low; those within the range of 0.05 to 0.01 are designated as medium; and those below 0.01 are assigned as high confidence. At least two peptide matches, each comprising a minimum of six amino acids in length, were configured for the protein identification process. To investigate low-abundance proteins, spectral clustering search nodes [42] were identified with the MS Amanda 2.0 [43] engine, as suggested by the developers. The spectrum clustering algorithm enhances the detectability of low-abundance proteins while simultaneously elevating the precision of the derived quantitative data without concomitantly increasing the noise of the datasets. Additionally, the msAmanda algorithm confidently provides more accurate spectra at the same FDR than Sequest on examined high-mass accuracy datasets. The combination of these nodes enabled the analysis of proteins that are low-abundant (FDR above 0.05, FDR class denotes low) but can demonstrate a notable fold change through label-free quantification (LFQ). LFQ was employed for the differential protein analysis using the Minora detection and precursor ion quantifier nodes of PD. The statistical significance of peptide abundances between groups was determined by one-way ANOVA using Benjamini–Hochberg correction. Proteins with a fold change ≥ 2 and an adjusted *p*-value of ≤0.05 were considered significant. The scrambled POMC secretome was selected as the control since no significant difference was observed when it was compared to the empty pCMV-3tag6 overexpressing the POMC secretome. Proteins exhibiting statistically significant ER stress-mediated differences (THA versus DMSO) in experimental groups with distinct genetic profiles for SIK2 (Control, SIK2 KD, and SIK2 WT) were evaluated. Furthermore, differences contingent on SIK2 manipulation (e.g., SIK2 KD versus Control) in ER stress or non-stress conditions were also calculated. The LFQ-based relative abundance values are listed in Table 2 and Appendix A.

The heat maps were constructed for the purpose of illustrating hierarchical clusters with a comparative analysis, utilizing PD, with the distance function set to “Euclidean”, which computes the geometric distance between two data points in multidimensional space. The linkage method was set to “Complete”, which calculates the dissimilarity between two clusters as the maximum distance between any two data points in different clusters (farthest neighbors). Gene Ontology (GO) term and biological pathway enrichment analyses were conducted using the ShinyGO software (v.0.80) [37] with the following parameters: the database was set to GO terms (biological process, molecular function, and cellular component), and Kyoto Encyclopedia of Genes and Genomes (KEGG), Reactome, and WikiPathways were designated as the source of terms and pathways, respectively. Additionally, the pathway size was set to a minimum of 2 and a maximum of 5000 genes. ClueGO (v.2.5.10) software, a Cytoscape plug-in [44], was used to determine and functionally visualize the nonsuperfluous biological terms for the resulting hierarchical clusters of genes in a functionally grouped network. The selection threshold was set to “medium”, i.e., at least 3 genes from the uploaded list, and 4% of genes selected for gene ontology terms were found to be associated with a term by referencing the GO term and pathway databases illustrated in the corresponding figure. The shared genes are also displayed as functional groups of terms and pathways based on a kappa score of 0.4, allowing detailed pathway analysis using the CluePedia extension. The minimum *p*-values were calculated using the FDR method derived from the hypergeometric distribution and Bonferroni step-down statistical options. STRING (v.3.7.6) [45] was used to visualize a network of protein–protein interactions in functionally related groups concerning experimental or database-predicted information. All comparative analysis cut-off was adjusted to 0.05 for *p*-value.

Immunoblotting and cell viability experiments were also carried out with samples from at least three independent biological replicates and technical triplicates (*n* = 9). Western blot quantification was conducted using the ImageJ software (v.1.48, USA) for densitometric measurements. All statistical analyses were performed using GraphPad Prism software (v.9.3, USA). One-way ANOVAs were conducted to compare the control and treated/manipulated groups, using Bonferroni post-tests for multiple comparisons. The resulting data are presented as mean ± standard deviation. The statistical parameters are provided in the figures and figure legends. In the process of writing the article, Scite.ai [46], an artificial intelligence-based program, was employed for the current literature review, particularly for the introduction section.

## 3. Results

### 3.1. Establishment of the POMC Model for Comparative Secretome Analyses

POMC neurons are the primary target of hypothalamic inflammation in diet-induced obese animal models and human subjects. Chronic exposure to dietary saturated fatty acids (SFAs) in these model organisms results in persistent ER stress and diminished autophagy flux in POMC neurons, which are repeatedly associated with neuronal loss and/or the development of leptin and insulin resistance. Taken together, the early activation of SIK2 in ER-stressed POMCs and its positive regulatory roles in ERAD and autophagy encouraged us to determine whether the regulatory role of SIK2 in these biological processes, which are tightly associated with the vesicular secretion machinery, has any effect on the cell secretome. To gain further mechanistic insight into the effect of SIK2 on organismal energy homeostasis and feeding behavior, we sought to identify secreted proteins of ER-stressed POMCs, the most salient feature of hypothalamic obesity. To achieve this goal, we designed an LC-MS/MS-based differential secretome study of the conditioned media of ER-stressed POMCs lacking or overexpressing SIK2. Prior to the start of comparative secretomic studies, the efficiency of SIK2 knockdown and overexpression in POMCs and their responsiveness to thapsigargin were confirmed by immunoblotting using specific antibodies (Figure 1A,C). Each shRNA suppressed SIK2 expression by approximately 70–80%SIK2-KD POMCs containing Exon2 were selected for further experiments. SIK2 appears to be a highly sensitive ER stress sensor because it responded instantaneously to thapsigargin treatment by reaching its maxima in 15 min and gradually decreasing below the control level within 3 h despite persistent stress induction and remaining relatively unchanged thereafter (Figure 1B). For the secretomics studies, cultured POMC cells with either endogenous, over-, or underexpressed SIK2 were exposed to thapsigargin for 6 or 12 h in the absence of serum. Secretomes subjected to 24 h of ER stress induction were excluded from the data analyses due to the reduced proliferation and viability of the neurons (Appendix A). The percentage of cell viability in serum-free media was found to be well above 90%, a value comparable to that observed in cells grown in complete media. The difference between these two values was found to be statistically non-significant. (Appendix A). Also, a notable effect on cell viability or proliferation was not observed with SIK2 manipulation. The respective secretomes were collected from the conditioned media via serial centrifugation. For the confirmation of secretome purity, immunoblotting for β-actin and α-tubulin cytoplasmic proteins was performed on secretome samples and full proteomes. A small amount of β-actin was detected in the secretome samples compared to the total lysate (Figure 1D). Although a 12 h time window is a commonly used interval that allows cells to synthesize their de novo secretomes, we also collected secretomes at 6 h post-ER stress induction to more accurately represent changes attributable to SIK2 activity (peak level in the initial 30 min) in the resulting secretome (Figure 1B). Furthermore, corroborating reports have indicated that cultured cells can produce a secretome effectively and at sufficient levels within 6 h following treatment with pharmacological agents [47].

### 3.2. LC-MS/MS-Based Analysis of POMC Secretomes

The secretome samples were analyzed by one-dimensional gel electrophoresis (1DE) and SYPRO-Ruby staining prior to mass spectrometry analysis. As anticipated, the buildup effect on the total secretome was noticeable at 12 h of ER stress induction. The differential expression of some secreted proteins could even be spotted on the gel by the naked eye (Appendix A). We detected 1306 total proteins that matched the *Mus musculus* protein databank with an FDR above 0.05 and a confidence interval ranging from medium to high. Among these proteins, 149 were soluble secreted proteins, 12 were shed proteins, and the rest were intracellular proteins. To detect low-abundance proteins, we used msAmanda and spectral clustering, which led to the identification of additional proteins, resulting in a final list of 885 soluble and secreted and 144 shed proteins (Figure 2A). Based on the LFQ abundance intensities, a heatmap was constructed, and subsequently, we applied clustering methodology to these 1029 proteins whose secretion were either increased or decreased together with respect to the SIK2 and ER stress variables. These hierarchical clusters of the POMCs secretome presented on the heatmap indicate differentially regulated proteins affected by SIK2 availability in the background of 6 and 12 h of ER stress conditions to differentiate the early and late response proteins relative to control in the secretome, respectively (Figure 2B,C). In the absence of ER stress, SIK2 overexpression produced the most significant (≥2-fold) increase in a group of related proteins (cluster 3) compared to SIK2 knockdown or control cells (Figure 2B). However, in response to 6 h of ER stress induction, significant increases of at least two-fold in the secreted proteins listed in clusters 5 and 6 were detected in SIK2 knockdown cells compared to both control and SIK2 overexpression. That is, ER stress-mediated SIK2 activation in POMCs suppresses the level of these proteins, especially during the early phase of ER stress response (Figure 2C). The comparative analyses of heatmap clusters revealed that SIK2 plays a pivotal role in regulating the secretion of proteins, particularly in the extracellular matrix, immune response, axon guidance and neurite outgrowth, and feeding behavior-related terms and pathways. These findings were exclusively observed in the early phase (6 h) of the ER stress response and appeared to be discharged from this duty by the implementation of late (12 h) ER stress. Although the observed fluctuations between the 6 h and 12 h time points in the control group in the absence of stress may be attributed to the potential effects of serum starvation, there are no statistically significant differences calculated according to LFQ-based analysis. On the other hand, the data showed that the proportion of differentially secreted proteins showing an increase at the 6 h time point was greater than that observed at the 12 h time point for both SIK2 knockdown and overexpressing cells compared to the control (Appendix A). For example, SIK2 overexpression produced a broader (~70%) yet contrasting incremental effect on these secreted proteins in clusters under no-stress conditions (Appendix A). Taken together, the data presented suggest that the effect of SIK2 on the POMC secretome is limited to the early phase of the ER stress response and is manifested by its ability to suppress various biological processes.

### 3.3. Functional Enrichment of the SIK2 and ER Stress-Derived Secretome

Functional enrichment analyses were performed on gene lists obtained from the hierarchically clustered proteins (Cluster 1–6) of the heatmap with respect to the SIK2 and ER stress variables using ShinyGO for analysis and identification of functional pathways and biological processes with the highest confidence (FDR < 0.05) (Figure 2B,C). In parallel, four clusters (Cluster 2 and 3, and Cluster 4 and 5) exhibit the SIK2 effect with greater discernibility on the heat map and have been assigned lower FDR values by ShinyGO and were transferred to ClueGO, a Cytoscape plug-in for further analysis. The nonredundant biological terms in the functionally grouped networks were evaluated using the Gene Ontology (Biological Process, Molecular Function, and Cellular Component), KEGG, Reactome, and WikiPathways databases of ClueGo (Figure 3 and Figure 4).

The majority of secreted proteins are associated with cellular processes, including the cellular response to external stimuli and biological regulation of the extracellular matrix. However, the percentage distribution of secreted proteins among the functionally related groups seemed to be significantly affected by SIK2 manipulations under physiological and ER stress conditions (Figure 3A and Figure 4A). In agreement with its systemic knockout phenotype, anti-inflammatory effects on macrophages, and cell survival effects on neurons and microglia, the effects of SIK2 on the identity and quantity of secreted proteins revealed the enrichment of functionally associated neuropeptides related to feeding behavior, cellular stress, and immune response, as well as neuron maturation and axon guidance in the absence of ER stress (Figure 3B). Furthermore, the secretion of chemotactic, inflammatory, appetite, and protein folding-associated proteins was altered by SIK2 under ER stress conditions (Figure 4B). Based on the LFQ abundance ratios, we identified 35 SIK2-regulated secreted proteins from the networks that are associated with obesity and feeding behavior, inflammation, and axon guidance and neurite outgrowth pathways (see Table 2 and Appendix A for the complete lists).

Upon analysis of these proteins, interestingly, during the early phase of the ER stress response (6 h), a group of proteins related to feeding and energy homeostasis, including growth/differentiation factor 15 (Gdf15), Serpinc1, amyloid precursor protein (App), and TIMP metallopeptidase inhibitor 1 and 2 (Timp1 and Timp2), were significantly increased in the secretome of SIK2-knockdown neurons. Surprisingly, a similar effect was reproduced, albeit to a slightly greater extent, by overexpressing SIK2 only in the absence of ER stress induction. This finding is compatible with the idea that SIK2 controls the level of these POMC-derived neuropeptides to communicate with cells in their immediate surroundings and with secondary neurons in the feeding circuitry under both physiological and early ER stress conditions. Conversely, one of the members of this group, calreticulin (Calr), decreased in SIK2-deficient POMCs in response to early ER stress induction, and this effect was reversed upon SIK2 overexpression, indicating differential regulation of counteracting factors within the same group (Table 2, see the obesity and feeding behavior group, red colored).

In addition, the secretion of axon guidance-related proteins, namely, perlecan (Hspg2), dickkopf-like protein 1 (Dkkl1), cathepsin B (Ctsb), matrilin-4 (Matn4), and nidogen-1 (Nid1), were induced collectively in the secretome of SIK2-depleted cells in response to early ER stress induction, and similar to SIK2 overexpression in feeding and obesity-related group, SIK2 overexpression increased the secretion of these proteins in a stress-independent manner, except for Dkkl1. In contrast to the general trend of other members of the group, the Dkkl1 level decreased with SIK2 overexpression during the early ER stress response (Table 2, see the axon guidance and neurite outgrowth group, green colored).

The third group of secreted proteins, which are tightly controlled by SIK2 under physiological and early ER stress conditions, belongs to the proinflammatory cytokine and chemokine family. In fact, following early ER stress induction, the expression profiles of members of this inflammatory gene group were similar to those previously mentioned. The levels of macrophage migration inhibitory factor (Mif), annexin A1 (Anxa1), fraktalkine (Cx3cl1), and granulin (Grn) in this inflammatory group were consistently and significantly upregulated in secretome POMCs lacking SIK2 upon early ER stress induction. As noted for the members of the former two groups, SIK2 had a positive effect on the levels of three inflammatory secreted proteins, Mif, Grn, and Cx3cl1, in the absence of ER stress but negatively regulated Anxa1 under early ER stress conditions (Table 2, see the inflammation group, blue colored).

As a general trend, the levels of all these proteins listed in the three groups are confined to the early phase of the ER stress response by stress-activated SIK2, and its deactivation during the late ER stress period releases these molecules, which apparently marks the end of the cell survival period (Appendix A). Furthermore, functional enrichment analysis of differentially secreted proteins grouped into clusters 2 and 4 (Figure 2B,C) revealed a concerted effort to activate the major survival-related PI3K-AKT pathway during the early phase of the ER stress response [48] (Appendix A), whereas death-related C/EBP homologous protein (CHOP) signaling emerged as the activity of SIK2 decreased in response to prolonged ER stress induction [49].

### 3.4. Protein–Protein Interaction Network Analyses

A list of POMC-derived secreted proteins regulated by SIK2 during the early phase of the ER stress response was directly uploaded to STRING software (v.3.7.6) for protein–protein interaction (PPI) network analysis (Table 2). A network of PPIs among functionally related proteins was visualized along with their functional annotations with reference to experimental or database-predicted information. STRING analysis revealed both inter- and intraconnections between the members of the three distinct gene groups obtained from the functional enrichment analyses, and the genes depicted in the PPI network are listed in the table below (Figure 5). Notably, PPI network analyses revealed an intricate interaction network between the members of three functionally related groups, accounting for their coregulation by SIK2, presumably downstream of the unfolded protein response (UPR), in ER-stressed POMC neurons. For example, the stress alarmone and anorectic protein Gdf15 interact with members of two other groups, including inflammatory Cx3cl1, axon guidance-related protein (Bmp3), and the obesity-related proteins Timp1 and erythropoietin (Epo). Concerning Timp1, a shed protein implicated in the inflammatory response plays a central regulatory role at the crossroads of protein–protein interactions between obesity-related proteins and feeding behavior-related proteins (Gdf15, Cd44, Sparc, and Angptl4) as well as axon guidance-related proteins (Ctsb, Hspg2, Col1a1, and 2) and neuroinflammation-related proteins (Cx3cl1, Anxa1, and Mif). We also found that cathepsin B (Ctsb), a lysosomal cysteine protease involved in protein turnover and proteolytic processing of amyloid precursor protein (APP), interacted with obesity- and feeding behavior-related proteins Calr, App, Hsp90ab1, Timp1, and Timp2 and inflammation-related Hmgb1 and Grn (Figure 5). Although APP is predominantly linked to the pathophysiology of Alzheimer’s disease, a recent report pointed out its association with obesity phenotypes such as insulin resistance and inflammation, which are regulated by tumor necrosis factor α (TNF-α) [50,51]. We provided an extended and targeted evaluation of these PPIs in the discussion.

## 4. Discussion

Based on the ER stress-mediated activation and localization of SIK2 to lipid rafts, we hypothesized that the effect of SIK2 on the POMC secretome could be readily delineated by probing qualitative and quantitative changes in secreted proteins via high-resolution LC-MS/MS-based secretome studies. To do so, we designed a comparative secretomic study to decipher the effects of SIK2 on the POMC secretome alone or in combination with pharmacologically induced ER stress conditions. Comparative analyses of the POMC secretomes specific to the early and prolonged phases of ER stress responses revealed the intriguing ability of SIK2 to control the expression profiles of dozens of secreted proteins concertedly only during the early phase of the ER stress response, consistent with its ER stress-dependent transient activation profile (Figure 1B). The functional clustering of these genes revealed six biologically relevant clusters, of which we focused on the four clusters (Cluster 2 and 3, and Cluster 4 and 5) related to obesity and feeding behavior, axon guidance and neurite outgrowth, and neuroinflammation based on their FDR values and significant overlap with the biological functions of SIK2 based on its knockout mice and modified cell lines, respectively. Our comparative secretome analyses demonstrated that the effect of SIK2 on the POMC secretome is limited to the early phase of the ER stress response. However, prolonged ER stress uncouples SIK2 from its effectors, thereby unleashing inflammatory and apoptotic secreted mediators to execute their functions within or on distantly located neuronal and non-neuronal cells. That is, during prolonged ER stress response, SIK2 repression of these secreted proteins, which are clustered in three different but functionally integrated groups, is relieved coincident with the deactivation of SIK2. As noted before, SIK2 suppresses the inflammatory content of the POMC secretome primarily in the early phase of the ER stress response, parallel to its effect on two other groups, including obesity and feeding behavior and axon guidance and neurite outgrowth (Table 2).

The early phase-specific profile imposed by SIK2 availability is commonly observed with the members of the three collectively acting groups. To avoid profile redundancy, we intentionally concentrated on a few representative secreted proteins selected from these three major groups (Figure 5 and Figure 6). One of the most interesting proteins listed in the table related to obesity and feeding behavior is Gdf15. It is an anorexigenic protein and acts through its cognate receptor, GFRAL, on the paraventricular nucleus (PVN) downstream of POMC neurons to end feeding and increase energy expenditure [52]. In our study, the level of Gdf15 in the SIK2 knockdown POMC secretome significantly increased in response to early ER stress induction and returned to the control level in response to SIK2 overexpression. However, without ER stress, the overexpression of SIK2 caused an increase in the Gdf15 level comparable to or even greater than that induced by ER stress in the absence of SIK2. It is commonly observed that the level of circulating Gdf15 is elevated in obese human subjects despite increased food intake [53]. Since exogenous Gdf15 injection ameliorated the obesity phenotype in obese animal models [54], therapeutic strategies targeting Gdf15 or its upstream regulator SIK2 in POMC neurons of the obese human brain will require the identification of differences in the Gdf15-GFRAL axis between the two species. In this group, calreticulin (Calr), which responds to early ER stress induction, exhibited a two-fold increase in the secretome of SIK2-overexpressing neurons compared to that of control neurons. Accordingly, the level of Calr decreased significantly with SIK2 deficiency independent of ER stress, in stark contrast to other members of the same group, excluding the coregulated proteins of other groups. Calreticulin is a multifunctional ER chaperone implicated in obesity and related metabolic disorders, and its secreted form acts as a molecular chaperone for App [55,56]. It has been implicated in several aspects of obesity and feeding behavior, in addition to its well-known role in the pathophysiology of Alzheimer’s disease [50]. We presumed that the reduced low level of Calr detected in the secretome of SIK2-deficient POMC neurons during the early ER stress response may interfere with the proper folding of App in relation to the pathogenesis of obesity and related diseases.

Cd44 antigen is a transmembrane protein expressed in both glial and neuronal cells, and its depletion has been linked to inflammation, neurite outgrowth, insulin resistance, and resistance to high-fat diet-induced obesity [57,58,59]. The level of Cd44 in the POMC secretome undergoes dynamic changes with respect to ER stress and SIK2 availability, which is in complete agreement with the findings of the gatekeeper model imposed by the absence of SIK2 during the early response phase of ER stress induction. However, SIK2 overexpression in POMCs caused a significant decrease in the Cd44 level, contrary to otherwise coregulated members of this and other related groups, especially those with no ER stress. Collectively, these results substantiated the anti-inflammatory and regulatory role of SIK2 in axonal guidance by decreasing the level of Cd44 in the secretome, thereby aggravating disease pathogenesis in its absence. Perlecan, a heparan sulfate proteoglycan, has been demonstrated to play a role in neuronal development and neurite outgrowth, with evidence supporting its involvement in brain repair following injury [60]. The level of perlecan in the SIK2 knockdown POMC secretome significantly increased in response to early ER stress induction and decreased in response to SIK2 overexpression. However, without ER stress, the overexpression of SIK2 caused an increase in the perlecan level comparable to that induced by ER stress in the absence of SIK2.

Fractalkine/Cx3cl1, a selected member of the inflammatory group, responds collectively with other members of the group to the gatekeeper molecule SIK2, which is a chemokine that plays a critical role in modulating neuronal–microglial communication [61,62]. It has a dual role in neuroinflammation, inhibiting the production of proinflammatory cytokines by microglia while acting as a chemoattractant for immune cells [63,64]. In agreement with the proposed model for the role of SIK2 in dictating early phase ER stress responsiveness of the POMC secretome, the Cx3cl1 level was increased 2–2.5-fold in POMC knockdown cells, and SIK2 overexpression reversed this increase in the early phase of the ER stress response. However, in the absence of ER stress, SIK2 overexpression increased the level of Cx3cl1, contrary to its absence. This indicates that the ability of SIK2 to suppress fractalkine levels in the POMC secretome during the early phase of the ER stress response is dependent on its UPR-mediated activation. Nevertheless, in the absence of ER stress, SIK2 is inactive and may act as a dominant negative to block endogenous SIK2 activity due to the tonic physiological level of ER stress. Granulin, a marker of chronic inflammation associated with insulin resistance, obesity, and type 2 diabetes mellitus, and its expression profile in the POMC secretome also obeyed the regulatory constraint imposed by a gatekeeper function of SIK2 [65]. For the extended coverage of the quantitative changes in the members of the three major groups, readers are referred to Table 2.

Overall, we propose a gatekeeper model for SIK2 because of its ability to maintain the transient control of these otherwise pathologically relevant inflammatory and apoptotic pathways during the early phase of the response to ER stress induction and to provide cells with a window of opportunity to rectify ER stress and maintain organismal homeostasis and cellular survival. In other words, SIK2 acts as a stopcock for the uncontrolled surge of double-edged biological processes, such as inflammation and autophagy, at the physiological level to maintain neuronal functions and survival. In contrast, prolonged unresolved ER stress tips the balance toward pathologically relevant chronic inflammation and interferes with autophagy by dampening the activity of SIK2, which agrees well with our data-driven gatekeeper model depicted in Figure 6. Although the molecular mechanisms responsible for the SIK2-mediated changes in the levels of secreted proteins are not addressed here, we propose that SIK2 could achieve this indirectly by regulating ERAD and autophagy processes or directly by modulating the autophagy-related unconventional secretion pathway. The effect of SIK2 on POMC secretome profiles can also be mediated at the transcriptional level by SIK2 modulating the activity of its downstream coactivators, CREB-regulated transcription coactivators (CRTCs), p300/CREB-binding protein, and transcription factor EB (TFEB), respectively [17,66]. However, it is equally or even more likely that SIK2 participates in the coat protein complex II (COPII) vesicle formation and/or cargo selection at the specified ER subdomains rich in lipid rafts [24]. Our laboratory and others reported direct and indirect physical interactions between SIK2 and the outer coat protein of COPII vesicles in immunoprecipitation-based mass spectrometry (IP-MS)-driven protein–protein interaction data (unpublished data). Based on the localization of SIK2 to lipid rafts on the ER membrane where COPII vesicles form, we speculated that SIK2 in lipid rafts could regulate either the formation of secretion vesicles or the selection of cargo proteins for secretion [22,24].

## 5. Conclusions

The novel aspect of the proposed work lies in undertaking the issue of obesity pathogenesis by examining the secreted neuropeptides and neurotransmitters of pathologically relevant POMC neurons, the central control unit for organismal energy homeostasis and energy expenditure. In diet-induced obese (DIO) animal models and human subjects, POMC neurons are either malfunctioning (resistant to anorectic hormones such as leptin and insulin) or depleted due to apoptotic loss. Therefore, the maintenance of healthy communication with their immediate neighbors (i.e., microglia, astrocytes, etc.) and distantly localized secondary neurons in the feeding circuitry (PVN) through their secreted proteins is vital for POMCs and, thereby, the life of an organism. We proposed an unprecedented regulatory role for SIK2 in the homeostatic control of obesity-related secreted proteins within physiological boundaries in response to the early phase of ER stress conditions. However, if unmitigated, persistent ER stress will unlock the SIK2 gate to allow a sudden increase in the levels of these obesogenic genes, which is compatible with disease pathogenesis. As noted in our model (Figure 6), when SIK2 was overexpressed, SIK2 could reverse this inflammatory state, although additional studies are needed to determine the effective dose for enhancing SIK2 activity due to the possibility of eradicating the basal response to ER stress.

In this study, we identified a number of POMC-driven pathologically relevant secreted molecules that we believe could be targeted individually or in combination to combat obesity and related diseases. It would be quite interesting and informative to obtain molecular insight into the possible regulatory roles of SIK2 both in the formation and cargo selection processes of secretory autophagosomes and COP II vesicles. Our comparative secretomic study revealed dozens of therapeutically valuable target proteins secreted by POMC neurons to modulate the immediate surroundings of neuronal and non-neuronal cells along distantly located secondary neurons of the PVN, all of which are implicated in the etiology and pathogenesis of obesity and various associated metabolic diseases, including neurodegeneration.

## Figures and Tables

**Figure 1 cells-13-01565-f001:**
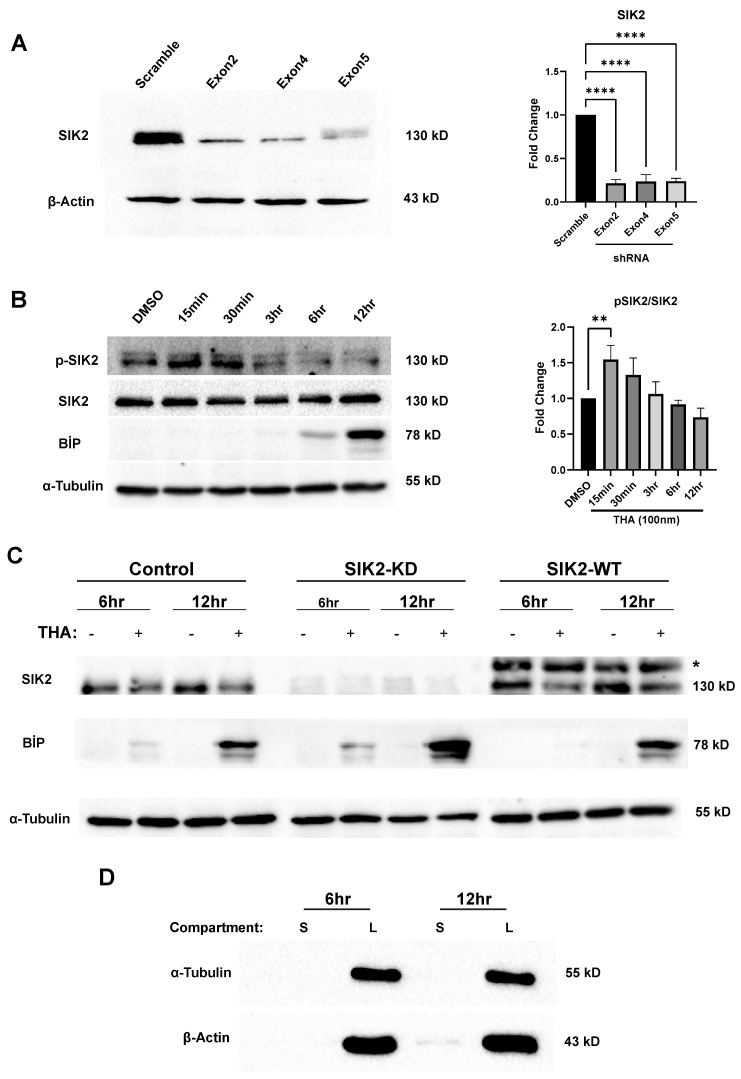
Confirmation of stable POMCs and secretomes. (**A**) Stable SIK2 KD POMC cell candidates carrying three different shRNAs were analyzed for SIK2 protein levels. The bar chart illustrates the expression level of SIK2 (*n* = 9). (**B**) SIK2 activation was confirmed via the expression of phospho-SIK2 (Thr175) and SIK2 under 100 nm THA at indicated time points. The bar chart illustrates the ratio of phospho-SIK2 to the total SIK2 (*n* = 9). (**C**) SIK2 manipulations were confirmed via immunoblotting on SIK2-knockdown (SIK2 KD), SIK2-overexpressing (SIK2 WT), and control POMC cells treated with 100 nm THA for 6 and 12 h. ER stress was also confirmed by BiP expression (*n* = 9). * Shifted band indicates 3xflag-tagged SIK2. (**D**) The purified secretomes at the indicated time points were evaluated for cytoplasmic contamination using antibodies specific for α-tubulin and β-actin. The secretomes (S) were compared to the cellular lysates (L) (*n* = 9). Error bars on densitometric graphs are means ± standard deviation (SD), for A: **** *p* < 0.0001, for B: ** *p* < 0.01.

**Figure 2 cells-13-01565-f002:**
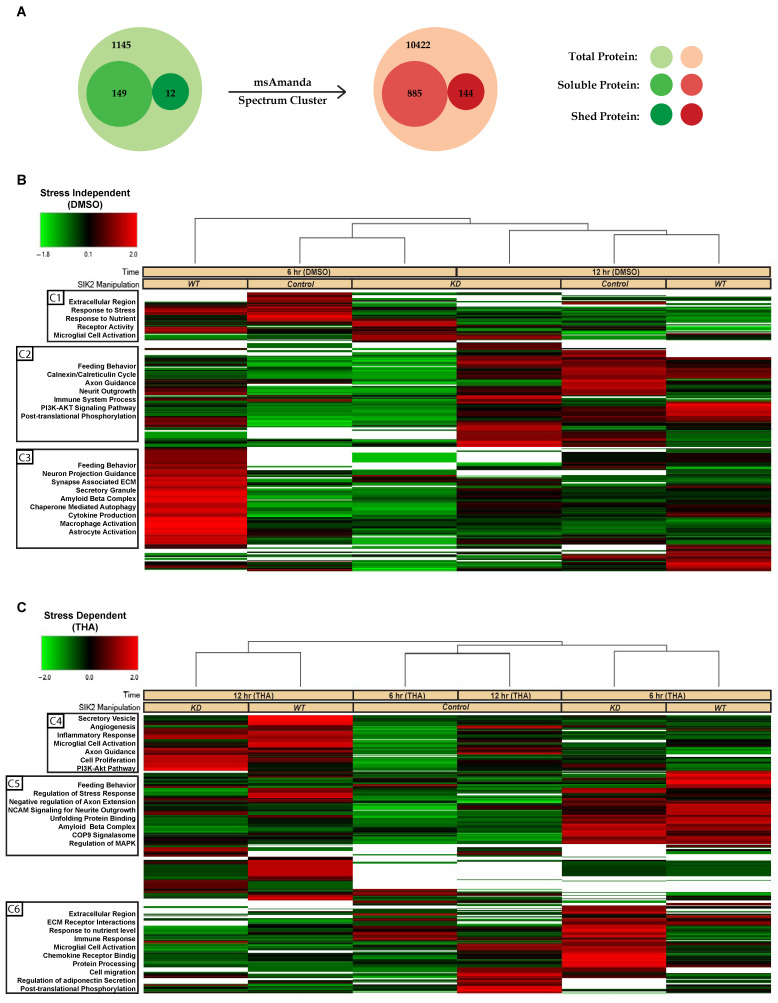
Visualization of the differentially secreted POMC proteins. (**A**) Soluble secreted and potentially shed proteins are indicated. The radius of the circles corresponds to the number of proteins. After further iteration using MS Amanda 2.0 and spectral clustering, the resulting protein counts are shown on the right side. Heatmap of POMC secretome proteins across SIK2 manipulations in the absence (**B**) and presence of ER stress (**C**) from hierarchical clustering using Proteome Discoverer 2.3. The rows represent the proteins, and the columns indicate groups of replicates. The ratio of protein abundances is represented by a color code indicating z-scores (green shows the decrease; black shows no change; red shows the increase), as illustrated in the accompanying legend. Functional annotation clustering with ShinyGO for the gene ontology terms/pathways is shown on the left, sorted by enrichment FDR score (<0.05).

**Figure 3 cells-13-01565-f003:**
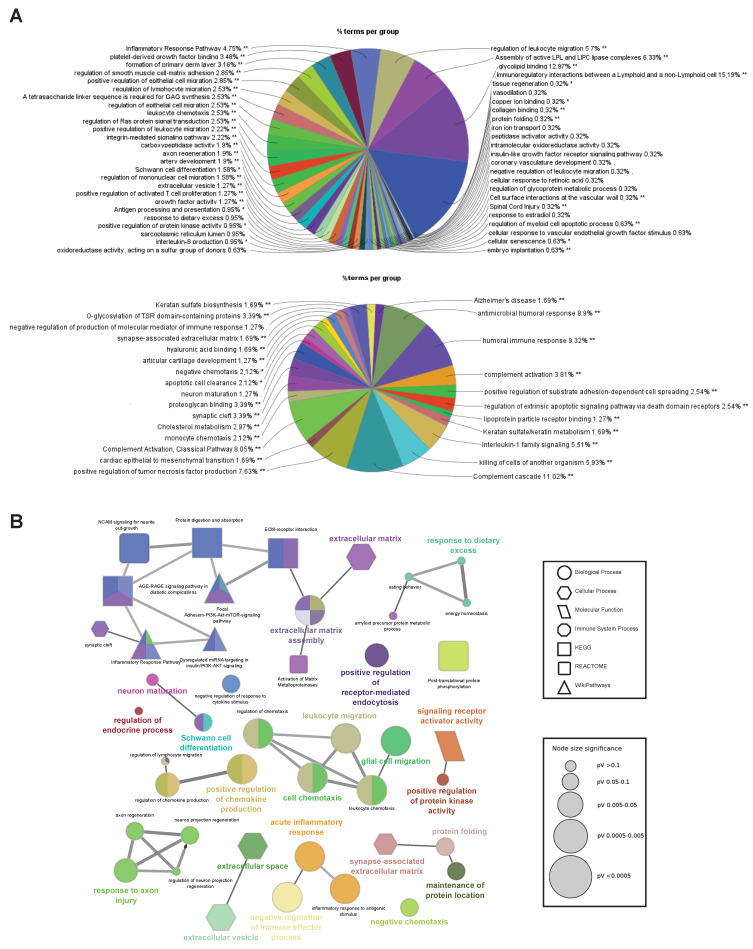
Stress-independent functional enrichment analysis of the SIK2-modified POMC secretome. (**A**) Pie charts generated by ClueGO and CluePedia showing the Gene Ontology terms and pathway analysis belonging to Cluster 2 (top) and Cluster 3 (bottom) with the percentage of terms/pathways per associated gene group. (**B**) Network analysis of selected functionally grouped terms/pathways. Each node represents a GO term/pathway, and the node size refers to the *p*-value, represented by a different node shape for each database (see legend for node size and shape). The edge represents the connection between the functionally related processes (according to kappa statistics). The different colors within the same node refer to the percentage of genes that were significantly enriched within the term belonging to the different groups. * *p* < 0.05, ** *p* < 0.01.

**Figure 4 cells-13-01565-f004:**
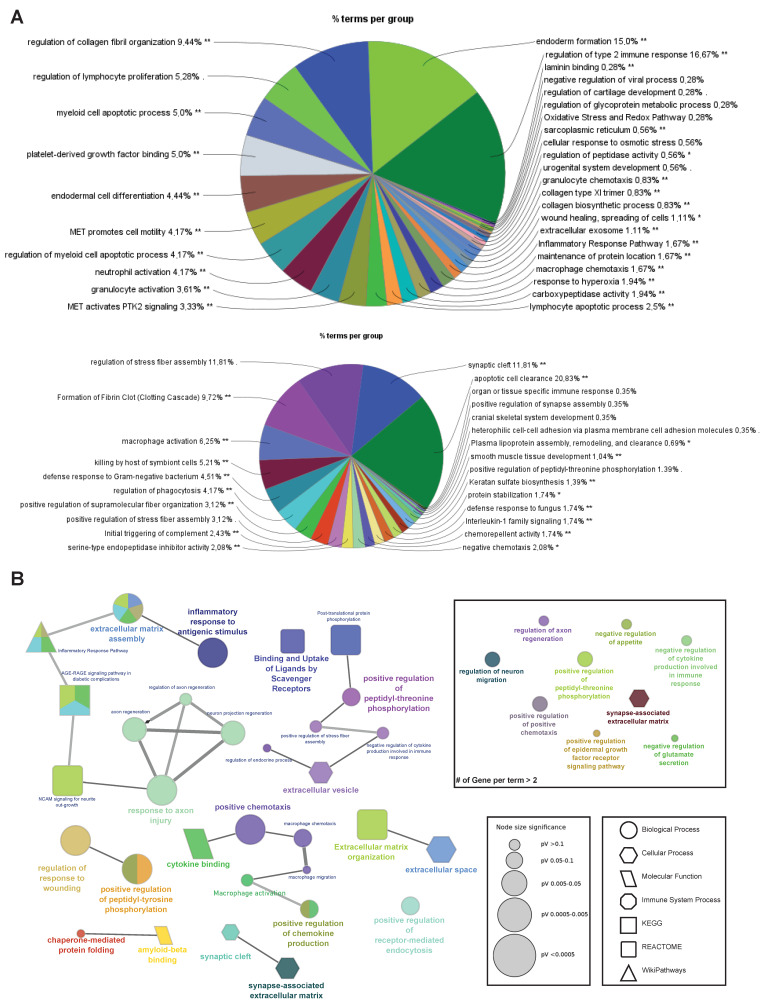
Stress-dependent functional enrichment analysis of the SIK2-modified POMC secretome. (**A**) Pie charts generated by ClueGO and CluePedia showing the Gene Ontology terms and pathway analysis belonging to Cluster 4 (top) and Cluster 5 (bottom) with the percentage of terms/pathways per associated gene group. (**B**) The network of selected functionally grouped terms/pathways. Each node represents a term/pathway, and the node size refers to the p-value, represented by a different node shape for each database (see legend for node size and shape). The edge represents the connection between the functionally related processes. The different colors within the same node refer to the percentage of genes that were significantly enriched within the term belonging to the different groups. The network was created by setting the criteria to have a term annotation for at least 2 genes instead of 3 genes (shown in the square on the right). * *p* < 0.05, ** *p* < 0.01.

**Figure 5 cells-13-01565-f005:**
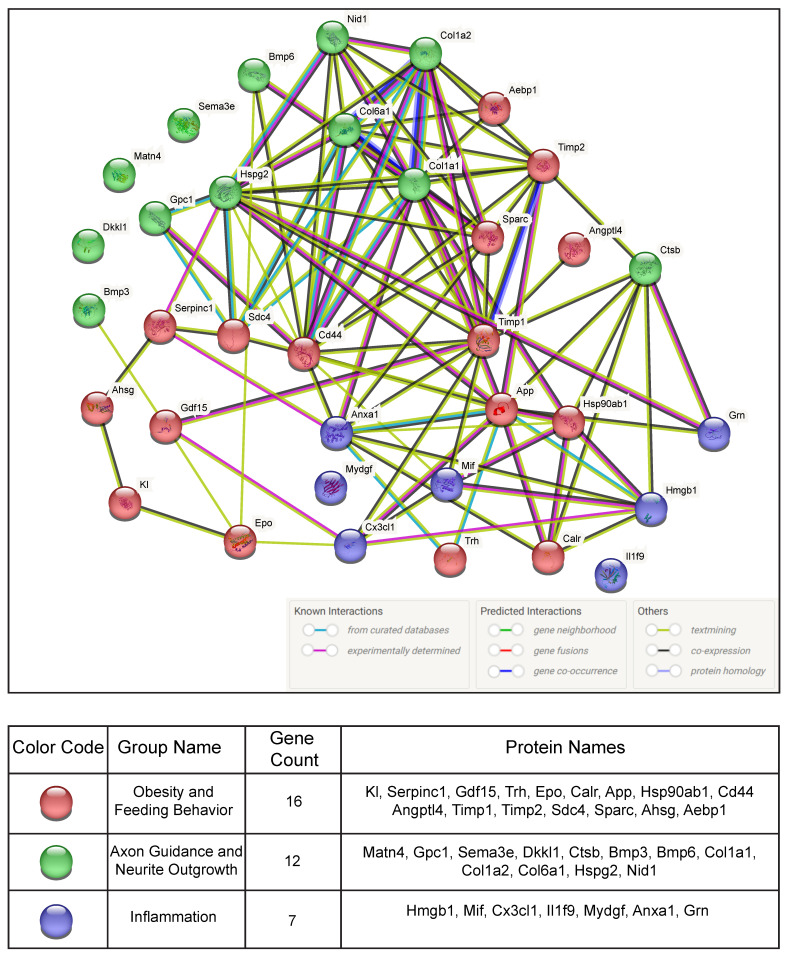
Illustration of protein–protein interaction network analysis using STRING. Proteins in the PPI network are shown as 3D spheres and labeled with their respective gene names, each representing a node in the network. The nodes are connected by edges representing their interaction. The color of the edges refers to their origin of interaction knowledge, as indicated. Proteins are clustered together according to their related biological terms, and each group is represented by a circle of different colors. The full names of the proteins are listed in Table 2.

**Figure 6 cells-13-01565-f006:**
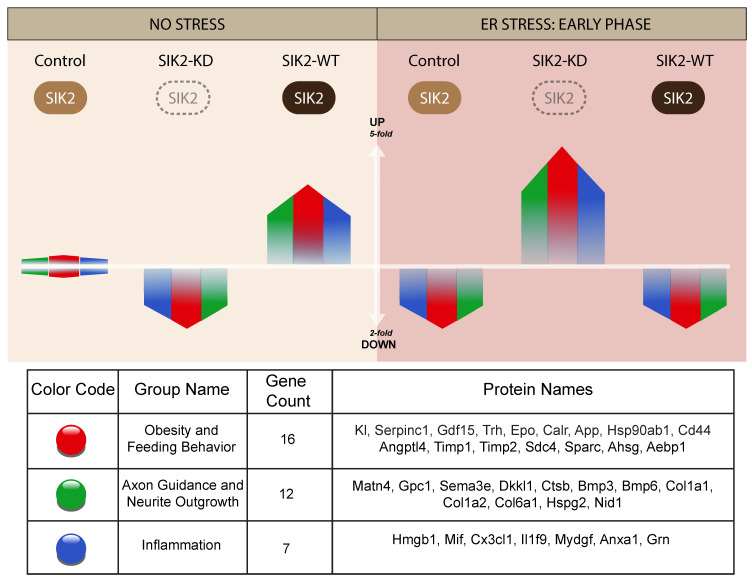
Differential and synergistic effects of SIK2 and ER stress on selected proteins of three major clusters. Relative fold changes are represented by colored bars according to their correlation with the differential gene expression (DGE) pattern. The bars illustrate the relative increase or decrease in secreted proteins. Protein abundance in all experimental conditions was normalized to the control group in the absence of stress. An extended list of relative fold changes for the genes excluded from the graph is provided (see Table 2).

**Table 1 cells-13-01565-t001:** Molecular cloning primers for the overexpression and shRNA systems.

Primer Name	Oligo Set
2F_KpnI (Tm: 61)	AAAAAGGTACCATGGTCATGGCGGATG
2R_KpnI (Tm: 64)	AAAAGGTACCCTAGGTCTCCCGGGCTAAG
mSIK2_Exon2_ shRNA_F	CCGGAATCTACCGAGAAGTACAGATCTCGAGATCTGTACTTCTCGGTAGATTTTTTTG
mSIK2_Exon2_ shRNA_R	AATTCAAAAAAATCTACCGAGAAGTACAGATCTCGAGATCTGTACTTCTCGGTAGATT
mSIK2_Exon4_ shRNA_F	CCGGAATTCTGTCTGCTGTTGATTACTCGAGTAATCAACAGCAGACAGAATTTTTTTG
mSIK2_Exon4_ ShRNA_R	AATTCAAAAAAATTCTGTCTGCTGTTGATTACTCGAGTAATCAACAGCAGACAGAATT
mSIK2_Exon5_ shRNA_F	CCGGAAGGACCACAGCTGGATATATCTCGAGATATATCCAGCTGTGGTCCTTTTTTTG
mSIK2_Exon5_ shRNA_R	AATTCAAAAAAAGGACCACAGCTGGATATATCTCGAGATATATCCAGCTGTGGTCCTT
Scrambled_ shRNA_F	CCGGCCTAAGGTTAAGTCGCCCTCGCTCGAGCGAGGGCGACTTAACCTTAGGTTTTTG
Scrambled_ shRNA_R	AATTCAAAAACCTAAGGTTAAGTCGCCCTCGCTCGAGCGAGGGCGACTTAACCTTAGG

**Table 2 cells-13-01565-t002:** SIK2 and early phase ER stress-dependent relative fold changes of three major clusters.

					Relative Abundance Ratio ^#^						
					6hr ER Stress Response (THA/DMSO) ^a^			6hr Stress Independent (DMSO)		6hr Stress Dependent (THA)	
	Accession ID	Protein Name (Abbreviation)	FDR Class *	HeatMap Cluster	Cntrl	KD	WT	KD/Cntrl ^b^	WT/Cntrl ^c^	KD/Cntrl ^b^	WT/Cntrl ^c^
**Obesity and Feeding Behavior**	**O35082**	Klotho (**Kl**)	Low	C1-C5	1.402	100	1.046	1.174	100	100	17.604
	**Q543J5**	Antithrombin-III (**SerpinC1**)	Medium	C1-C5	1.026	1.652	0.648	0.97	1.84	2.225	1.238
	**Q9Z0J7**	Growth/Differentiation Factor 15 (**Gdf15**)	Low	C2-C5	1.645	2.185	1.133	0.871	4.022	2.455	1.368
	**Q62361**	Pro-thyrotropin-releasing hormone (**Trh**)	Low	C2-C5	0.01	1.036	0.786	1	1	100	100
	**B7ZMY9**	Erythropoietin (**Epo**)	Low	C2-C4	18.99	0.01	12.678	9.936	1	0.066	0.969
	**B2MWM9**	Calreticulin (**Calr**)	High	C1-C5	4.119	0.468	11.924	0.45	0.917	0.178	3.571
	**P12023**	Amyloid-beta precursor protein (**App**)	Medium	C2-C6	0.995	2.27	0.706	0.498	3.189	3.072	1.518
	**P15379**	CD44 antigen (**Cd44**)	High	C1-C4	0.596	2.069	0.897	2.002	0.567	5.964	1.285
	**Q71LX8**	Heat shock protein HSP 90-beta (**Hsp90ab1**)	High	C3-C5	1.208	2.398	0.651	0.927	4.802	2.361	1.933
	**Q9Z1P8**	Angiopoietin-related protein 4 (**Angptl4**)	Low	C2-C5	1.126	1.136	1.238	7.973	12.277	7.305	4.612
	**P12032**	Metalloproteinase inhibitor 1 (**Timp1**)	High	C2-C4	0.127	0.36	0.073	0.373	4.778	1.626	2.862
	**P25785**	Metalloproteinase inhibitor 2 (**Timp2**)	High	C2-C5	0.611	1.144	0.276	0.733	4.885	1.374	1.454
	**O35988**	Syndecan-4 (**Sdc4**)	Medium	C1-C5	0.376	0.313	0.56	0.318	1.542	0.212	1.098
	**Q5NCU4**	SPARC (**Sparc**)	High	C2-C6	0.055	0.187	0.037	0.799	1.654	2.794	1.032
	**A0A338P7H5**	Alpha-2-HS-glycoprotein (**Ahsg**)	High	C3-C5	0.975	1.519	0.916	0.467	2.74	2.549	2.523
	**Q640N1**	Adipocyte enhancer-binding protein 1 (**Aebp1**)	High	C2-C4	0.01	0.01	0.021	0.711	14.224	0.01	9.171
**Axonal Guidance and Neurite Outgrowth**	**O89029**	Matrilin-4 (**Matn4**)	Low	C1-C6	0.767	2.938	0.788	0.804	1.616	2.458	1.914
	**Q9QZF2**	Glypican-1 (**Gpc1**)	Medium	C2-C6	0.01	0.01	0.345	0.01	1.181	0.01	12.076
	**P70275**	Semaphorin-3E (**Sema3e**)	Low	C2-C5	7.148	1.32	0.869	1.519	4.832	1.396	4.177
	**Q9QZL9**	Dickkopf-like protein1 (**Dkkl1**)	Low	C2-C4	0.452	4.363	0.01	6.27	1	17.097	0.01
	**P10605**	Cathepsin B (**Ctsb**)	High	C3-C6	0.128	0.283	0.079	0.754	1.737	1.884	1.095
	**A0A0G2JE75**	Bone morphogenetic protein 3 (**Bmp3**)	Low	C3-C5	0.984	8.54	1.566	0.348	0.731	3.47	3.098
	**P20722**	Bone morphogenetic protein 6 (**Bmp6**)	Low	C1-C6	1.142	3.508	0.708	1.929	0.406	0.872	0.359
	**P11087**	Collagen alpha-1(I) chain(**Col1a1**)	High	C2-C4	0.238	0.552	0.292	0.994	2.571	4.522	2.359
	**Q01149**	Collagen alpha-2(I) chain (**Col1a2**)	High	C2-C5	0.387	0.576	0.251	0.83	4.04	4.526	3.055
	**Q04857**	Collagen alpha-1(VI) chain (**Col6a1**)	High	C2-C6	0.836	2.229	0.604	0.329	0.994	1	0.891
	**B1B0C7**	Perlecan (heparan sulfate proteoglycan 2)(**Hspg2**)	High	C1-C6	0.287	1.066	0.103	0.673	1.529	2.135	0.598
	**P10493**	Nidogen-1 (**Nid1**)	High	C2-C4	0.121	0.342	0.104	0.593	3.745	3.005	2.665
**Inflammation**	**Q58EV5**	High mobility group protein B1 (**Hmgb1**)	Medium	C2-C5	1.027	1.904	0.787	0.633	1.679	3.271	1.944
	**P34884**	Macrophage migration inhibitory factor (**Mif**)	High	C2-C4	0.721	2.153	0.817	0.864	2.139	5.431	3.303
	**O35188**	Fractalkine (**Cx3cl1**)	Medium	C1-C6	0.892	2.532	0.596	0.549	1.766	2.396	1.016
	**Q8R460**	Interleukin-36 gamma (**Il36g**)	Medium	C1-C6	1.131	2.242	0.662	0.791	1.432	2.021	1.937
	**Q9CPT4**	Myeloid-derived growth factor (**Mydgf**)	High	C2-C4	29.652	24.199	26.057	1.423	1.431	1.28	1.348
	**Q4FJV4**	Annexin (**Anxa1**)	Medium	C2-C5	0.068	0.966	0.01	0.087	0.084	1.578	0.01
	**Q544Y8**	Granulin (**Grn**)	High	C2-C6	0.417	1.921	0.267	0.669	6.055	2.43	1.878

* FDR class, High: q value ≤ 0.01; Medium: q value ≤ 0.05; Low: q value > 0.05. # For proteins with a fold change ≤ 2: *P* Adj. ≤ 0.05. a: The abundance of secreted proteins in ER-stressed cells compared to that of unstressed cells. b: The abundance of secreted proteins in SIK2 knockdown cells compared to control cells. c: The abundance of secreted proteins in SIK2 overexpressing cells compared to control cells.

## Data Availability

Data are available upon request by qualified researchers to the corresponding author.

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
