# Peer review of "SIK2 Controls the Homeostatic Character of the POMC Secretome Acutely in Response to Pharmacological ER Stress Induction"

_cells, 2024, doi:10.3390/cells13181565_

Round 1
Reviewer 1 Report
Comments and Suggestions for Authors
The paper addresses how the composition of proteins secreted from a cell line derived from POMC neurons of the arcuate nucleus changes in response to reducing the Ca content of the endoplasmic reticulum and also in response to manipulating the expression level of a protease, SIK2. POMC is an important component of the brain network controlling food intake, which provides the significance of the data, which is properly explained. The secretome was analyzed with mass spectrometry, an appropriate technique for this purpose. There are, however, several flaws in the manuscript, which have to be addressed.
1. Most importantly, only n=3 samples were measured, which is not sufficient to conclude significance.
2. The differentially secreted genes must be reported in supplementary tables. Fold changes are not enough for these gene lists, rather, p values have to be shown as well.
3. It is not properly reported how confident we can be in the MS identification of proteins. It has to be claimed how many digested peptides were used for the identification of each protein.
4. It is not properly explained why the number of intracellular proteins is higher in the secretome than the number of secreted proteins. Furthermore, only a single value is provided for this ratio, so we do not know if it changes with the conditions.
5. The number of secreted proteins involved in the analysis must be provided for each single sample.
6. Why do the authors believe their findings are specific to POMC neurons? Information on the expression pattern of SIK2 in the brain specifically in the arcuate nucleus must be provided.
7. The effect of SIK2 overexpression and knockdown is novel information. Has ER impact of stress on the secretome of POMC cells been investigated in the past?
8. Are the ER stress-induced changes specific to the POMC neurons? Is there any previous literature data on the secretome of any other cell type to compare to?
9. Introduction to Parkinson’s disease is irrelevant.
10. In Fig. 1b, DMSO data are shown only once instead of all time points.
11. The discussion starts with introducing SIK2 intracellular pathways, which should be placed in the Introduction.
12. Discussion should contain a comparison of the results with previous literature data.
Comments on the Quality of English LanguageFairly good
Author Response
- Most importantly, only n=3 samples were measured, which is not sufficient to conclude significance.
All experiments, particularly those involving mass spectrometry, comprise three biological and three technical replicates, as delineated in lines 325-326. It is important to acknowledge that the requirement of replicate numbers may differ depending on the specific objective of identification or quantification in mass spectrometry study. For identification reasons, it is preferable to have at least three biological replicates instead of technical replicates. However, if the goal is to analyze the data statistically for quantification, additional technical replicates are included. For instance, in a recent study, the subproteomics analysis consisted of six samples, which included two biological replicates and three technical replicates (n=6) (doi:10.1038/s41467-019-13973-x). However, our study, which consisted of a sample size of 9, meets the statistical criteria for conducting LC-MS/MS-based proteomics studies, both in terms of quantitative and qualitative analysis, employing high-resolution Orbitrap mass spectrometry. In our study, the sample number is specified as n=3 indicating the number of biological triplicates only. In the immunoblotting and cell viability experiments, the number of technical replicates was not explicitly stated in line 385. To clarify this inconsistency, we corrected n as equal to 9 (n=9) in the material methods and figure legends which is in line with the conventions employed in the existing literature.
- The differentially secreted genes must be reported in supplementary tables. Fold changes are not enough for these gene lists, rather, p values have to be shown as well.
In our study, the fold changes between the experimental groups in the LFQ analysis were subjected to ANOVA using Benjamini-Hochberg and P-adjustment values were obtained for each comparison. The tables present the differentially secreted proteins that meet the criteria of a minimum 2-fold difference and a P Adj. ≤0.05 cutoff, and that are functionally related to each other and to the concept under investigation. As implicated in the Data Availability Statement, the entire list, including those outside the specified filter, can be made available upon request, given the extensive nature of the data set. However, it was observed that the LFQ-based statistical analysis was not properly explained in lines 349-351, which caused this confusion. The method has been elaborated upon, and the table legends have been augmented with P Adj.
- It is not properly reported how confident we can be in the MS identification of proteins. It has to be claimed how many digested peptides were used for the identification of each protein.
In line 333, it is indicated that within the context of protein identification, validations were conducted with the "Perculator" node, based on the FDR scores of the detected peptides. However, the number of peptide matches is not included, as pointed out by the referee. In our study, the peptide filter parameters were set to require a minimum of two peptide matches for a protein inference process. The text on MS identification and confidence information has been expanded in greater detail in lines 333-339.
- It is not properly explained why the number of intracellular proteins is higher in the secretome than the number of secreted proteins. Furthermore, only a single value is provided for this ratio, so we do not know if it changes with the conditions.
In secretome experiments, contamination of the medium by intracellular proteins is a common occurrence due to the lysis and damage of cells. Indeed, this phenomenon is commonplace in high-resolution LC-MS/MS-based secretomics studies, with an average of 12% secreted protein being detected in the conditioned medium, while the remainder is of cytoplasmic and intracellular origin (doi:10.1016/j.ab.2022.114846). Furthermore, it is unsurprising that intracellular proteins from dead cells, which represent a statistically insignificant 10% of the total, were detected by LC-MS/MS, as illustrated in Supplementary Figure S1.
- The number of secreted proteins involved in the analysis must be provided for each single sample.
The referees point is well taken. In the total number of secreted proteins (total secretome) detected in our study, our objective was to examine differentially secreted proteins within the functionally related groups. Proteins with a fold change below the 2-fold cutoff were deemed insignificant and excluded from the analysis. However, If needed the number of secreted proteins detected for each condition can be provided as an attachment.
- Why do the authors believe their findings are specific to POMC neurons? Information on the expression pattern of SIK2 in the brain specifically in the arcuate nucleus must be provided.
The Allen Brain Atlas indicates that SIK2 is expressed in the hypothalamus of the mouse brain by both RNA and in-situ hybridization-based methods (https://mouse.brain-map.org/experiment/show?id=71281740). As illustrated in Figures 1a and 1b, the immunoblotting method indicates the presence of a discernible level of SIK2 expression in POMC cells. Moreover, the systemic SIK2 knockout mouse demonstrated a POMC specific physiological phenotype marked with reducing feeding and energy expenditure (doi: 10.2337/db13-1423) as underlined in the abstract (lines 12-15), (https://www.informatics.jax.org/allele/genoview/MGI:5762871). Additionally, we also confirmed the expression of human SIK2 in hypothalamic arcuate nucleus (https://www.proteinatlas.org/ENSG00000170145-SIK2/brain/hypothalamus#rnaseqhypothalamus)
(https://www.proteinatlas.org/ENSG00000170145-SIK2/brain)
- The effect of SIK2 overexpression and knockdown is novel information. Has ER impact of stress on the secretome of POMC cells been investigated in the past?
No, to the best of my knowledge nobody has addressed this question although few studies looked at the effect of ER stress on secretome of non-neuronal cells as exemplified in the next question. As stated in the introduction (lines 114-119), it is well established that in diet-induced obesity ER stress, impaired autophagy and inflammation underlies the POMC dysfunction marked by insulin and leptin resistances and death. In fact, these in vivo findings along with the SIK2 knockout mouse phenotype formed the foundation of our hypothesis.
- Are the ER stress-induced changes specific to the POMC neurons? Is there any previous literature data on the secretome of any other cell type to compare to?
Based on the hypothalamic obesity model POMC neurons are the primary cell types affected by the diet induced inflammatory state in conjunction with ER stress and abrogated autophagy underlining the disease etiopathology. However, it would be interesting to investigate the effect of ER stress on microglia secretome in the arcuate nucleus of DIO mice since these cells are suggested to play role in hypothalamic inflammation as well. Unfortunately, there is no secretome studies with microglia or any other cell types of the arcuate nucleus to compare our data with. Beside, we do not only measure the effect of ER stress on POMC secretome but rather we focused on its major effector stress kinase SIK2 (downstream of UPR) and its effect on POMC secretome. What we found in the literature is very scarce and limited in providing useful data. There are few studies but they are conducted with non-neuronal cell types yet measuring the effect of only ER stress on their secretome (reference below). Therefore, there is no point to make any logical comparison with our experimental settings which utilizes physiologically relevant POMC neurons and suspected gene SIK2 in disease pathogenesis.
The effect of ER stress on myocyte secretome (DOI: 10.1016/j.yjmcc.2020.04.012).
- Introduction to Parkinson’s disease is irrelevant.
We have previously mentioned two specific diseases, namely Parkinson's disease and ALS, in the context of line 103. Actually, we attempted to indicate that the neuroprotective effects of secretome may be applicable to a wider range of neurodegenerative diseases. We understand your preference for removing this content, and we leaved it as a generalization, referring to neurodegenerative diseases without naming the specific disease in question at this point.
- In Fig. 1b, DMSO data are shown only once instead of all time points.
Our preliminary findings indicate that the DMSO at the given concentration (0.1%, v/v) does not elicit a response. Furthermore, it does not alter the phosphorylation status of SIK2 within the tested time intervals (0-12 hours). Should you require further details, we are willing to share our data. Furthermore, the results of a recent study indicate that the concentration in question is considerably below the threshold for cytotoxic effects of DMSO (0.5-1%) even when the treatment is extended to 12 hours (10.1016/j.brainresbull.2016.11.004).
- The discussion starts with introducing SIK2 intracellular pathways, which should be placed in the Introduction.
We totally agree with the referee on this and made the necessary changes accordingly, in the context of lines 83-99.
- Discussion should contain a comparison of the results with previous literature data.
We agree with the referee pointing out the necessity for including a section of comparison with the existing literature. However, our data provides an unprecedented data which is hardly comparable to existing research as noted in our answers to critics in 7 and 8. We noticed referee accidentally points out SIK2 as a protease instead of ER stress responsive kinase. Although irrelevant in experimental settings those few studies focusing exclusively on the effect of ER stress on the cellular secretome concertedly indicated that ER stress has a beneficial effect on cell survival in paracrine and autocrine manner. Although we also noted secreted factors contributing to the cell survival in the secretome of ER stress-induced control group (no SIK2 modification) Taken together there is no point to include this content limited and strategically remotely related study in our discussion.
Reviewer 2 Report
Comments and Suggestions for Authors
The authors tested the potential involvement of SIK2 (salt inducible kinase 2) in regulation of the energy metabolism in POMC (pro-opiomelanocortine) neurons, indicating SIK2 as a gatekeeper for the secreted inflammatory factors and apoptotic molecules while its lack associates the occurrence of proinflammatory and apoptotic signals with the dysregulation of POMC neurons. Although the study is of sufficient significance and originality, and has the preprint version online since 08.08.2024. (https://www.preprints.org/manuscript/202408.0611/v1, (doi: 10.20944/preprints202408.0611.v1), there are several issues that need to be addressed.
Concerns:
1. Full affiliations for all authors are necessary.
2. In the section Materials and methods, subsection 2.1. Cell Culture and Chemical Treatments, do references 32 and 33 refer to the statement: “It has been previously demonstrated that these cells secrete α-MSH and, consistent with their origin, respond to anorectic hormones such as insulin and leptin as well as the ER stressor palmitate.”? If not, provide appropriate reference(s). Moreover, for some abbreviations, including DMEM, DMSO, etc. the description should be provided.
Author Response
- Full affiliations for all authors are necessary.
The cover page of the manuscript indicated the full affiliations of all authors as 1 and 2.
- In the section Materials and methods, subsection 2.1. Cell Culture and Chemical Treatments, do references 32 and 33 refer to the statement: “It has been previously demonstrated that these cells secrete α-MSH and, consistent with their origin, respond to anorectic hormones such as insulin and leptin as well as the ER stressor palmitate.”? If not, provide appropriate reference(s). Moreover, for some abbreviations, including DMEM, DMSO, etc. the description should be provided.
In accordance with the updates made, the references cited are 34 and 35, rather than 32 and 33. We have confirmed that the references are relevant to the context in which they are cited. Thank you for pointing out to abbreviations that we mistakenly overlooked, in accordance with your suggestion, the necessary explanations have been added to the previously unspecified abbreviations.
Reviewer 3 Report
Comments and Suggestions for Authors
I have reviewed the article by Turkuner et al., titled SIK2 Controls the Homeostatic Character of POMC Secretome Acutely in Response to Pharmacological ER Stress Induction. I admire the present manuscript as it presents a comprehensive and well-executed study that significantly contributes to the understanding of the molecular mechanisms involved in hypothalamic obesity. This is a very important and crucial topic in the present world. The authors have also presented a thorough explanation of the role of SIK2 in regulating the secretome of POMC neurons under ER stress. This highlights its potential as a therapeutic target and hence could be used for drug discovery in future studies. I found the experimental design to be good. The data has been meticulously arrived at, and analyzed. The conclusions are well-supported by the findings presented. I must say that given the originality of the work, the depth of analysis, and the potential impact on the field of neurobiology and metabolic disorders, I recommend the acceptance of this manuscript for publication in its present form.
Author Response
We would like to express our gratitude for the insightful and motivating feedback on our study.
Round 2
Reviewer 1 Report
Comments and Suggestions for Authors
Thank you for properly responding to all the critiques.